# Wearable Biosensors: An Alternative and Practical Approach in Healthcare and Disease Monitoring

**DOI:** 10.3390/molecules26030748

**Published:** 2021-02-01

**Authors:** Atul Sharma, Mihaela Badea, Swapnil Tiwari, Jean Louis Marty

**Affiliations:** 1School of Chemistry, Monash University, Clayton, Melbourne, VIC 3800, Australia; 2Department of Pharmaceutical Chemistry, SGT College of Pharmacy, SGT University, Budhera, Gurugram, Haryana 122505, India; 3Fundamental, Prophylactic and Clinical Specialties Department, Faculty of Medicine, Transilvania University of Brasov, 500036 Brasov, Romania; mihaela.badea@unitbv.ro; 4School of Studies in Chemistry, Pt Ravishankar Shukla University, Raipur, CHATTISGARH 492010, India; swapnil.tiwari7@gmail.com; 5University of Perpignan via Domitia, 52 Avenue Paul Alduy, CEDEX 9, 66860 Perpignan, France

**Keywords:** wearable biosensor, healthcare monitoring, biomarkers, biological fluids, physiological conditions

## Abstract

With the increasing prevalence of growing population, aging and chronic diseases continuously rising healthcare costs, the healthcare system is undergoing a vital transformation from the traditional hospital-centered system to an individual-centered system. Since the 20th century, wearable sensors are becoming widespread in healthcare and biomedical monitoring systems, empowering continuous measurement of critical biomarkers for monitoring of the diseased condition and health, medical diagnostics and evaluation in biological fluids like saliva, blood, and sweat. Over the past few decades, the developments have been focused on electrochemical and optical biosensors, along with advances with the non-invasive monitoring of biomarkers, bacteria and hormones, etc. Wearable devices have evolved gradually with a mix of multiplexed biosensing, microfluidic sampling and transport systems integrated with flexible materials and body attachments for improved wearability and simplicity. These wearables hold promise and are capable of a higher understanding of the correlations between analyte concentrations within the blood or non-invasive biofluids and feedback to the patient, which is significantly important in timely diagnosis, treatment, and control of medical conditions. However, cohort validation studies and performance evaluation of wearable biosensors are needed to underpin their clinical acceptance. In the present review, we discuss the importance, features, types of wearables, challenges and applications of wearable devices for biological fluids for the prevention of diseased conditions and real-time monitoring of human health. Herein, we summarize the various wearable devices that are developed for healthcare monitoring and their future potential has been discussed in detail.

## 1. Introduction

Nowadays, the design and development of wearable biosensors with their potential in human health monitoring and personalized medicine have gathered significant attention. Wearable biosensors (WBSs) are portable electronic devices that integrate sensors into/or with the human body in the forms of tattoos [1], gloves [2], clothing [3] and implants [4], realizing in vivo sensing, data recording and calculation using mobile or portable devices. WBSs are known to create two-way feedback between physicians and patients [5]. These devices also enable the non-invasive and real-time quantification of various biochemical markers in the human body fluids such as saliva, sweat, skin, and tears [6]. With the novel innovation and advancements in material science as well as development in mechanical engineering and wireless communication technologies, various wearable devices (watch, bands, etc.) have been developed and employed for processing and simultaneously analyzing biomarkers to improve healthcare management [6,7]. With an ageing population, the evidence of food safety and disease outbreaks has increased. The market sale of wearable technology is expected to rise up to USD 70 billion by 2025 for their ease of use [3].

A typical biosensor is a composition of the two basic functional units, i.e., a “biorecognition element or bioreceptor” (enzyme, antibody, DNA, nucleic acid, peptide, etc.) and a physicochemical transducer such as optical, electrochemical, piezoelectric, and thermal (Figure 1). The bioreceptor is responsible for selectively recognizing the target analyte and the transducer is responsible for the conversion of a biorecognition event into a measurable signal [8,9]. Initially, the biosensing devices were designed and developed for in vitro or single-use measurements, for example, glucometer, glucose test strips, and glucowatches. Furthermore, the advancement in biosensor technologies has paved the way to commence improvements in modern wearable biosensors for non-invasive monitoring in healthcare and biomedical applications [10].

In wearable devices, the key component is wearable sensors. Furthermore, these wearable sensors with integrated functions of measuring identified markers solve various noticeable problems in the health, medical and sports field. Based on different parameters measured, WBSs are classified into motion state, biophysical and biochemical sensors [6]. To measure human physical parameters such as gait, sleep, tremor for real-time monitoring and collection of long-term information, the motion state sensors are mainly used [11,12]. With integrated lab-on-chip technology, the wearable biochemical sensors parallelly measure the trace and processing of different samples [13]. Scientists and lab workers can precisely measure biomarkers in biological fluids in order to monitor health conditions and metabolism utilizing wearable biochemical sensors [14,15,16]. Wearable biophysical sensors intrigued characteristics is to contact with skin to provide real-time measurement of biophysical parameters such as blood pressure, heart rate, and temperature that possess significant values in healthcare applications [17,18]. Among them, the biophysical and motion state sensors are available in the market and widely used by consumers. Whereas, the biochemical biosensors are still not marketized and possess significant potential as the biological fluids are complex matrices and challenging to detect the analyte of interest [19].

Integration of sensors for the detection of diversified biomarkers in the future of wearable biosensors is a challenge, and it requires a continuous breakthrough in sensing devices. Among various forms of biosensors, the electrochemical-based sensors show remarkable advantages of ease of construction, higher sensitivity, quick response, and ability to work with low consumption of power [20]. Sensing electrodes play a significant role in the constitution of wearable sensors, mainly based on the electrochemical method [21]. On the other hand, critical issues such as functional material and the manufacturing technologies used in the preparation of sensing electrodes still is a challenge and need to be addressed to improve the performance of wearable biochemical sensors [22].

The conventional sensing electrodes technology in wearable sensing are metal-based film electrodes. Various advancements have been reported in search of new materials such as hybrid and metallic nanoparticles, nanocomposite, carbon and polymeric materials to be used as the electrode materials in the construction of the wearable biochemical sensors resulted in the improvement of sensors performance [1,23,24,25,26]. On the other hand, the ingenious micro-manufacturing technologies afford durable and robust support for the design and optimizing working parameters of the sensing electrodes [6,27]. In recent years, significant efforts have been devoted to preparing such kinds of wearable sensors as more and more emphasis is to recognize the various biomarkers that affect health [28]. Several excellent reviews describing wearable biosensors have been reported. Multiple perspectives have already been reported [29,30,31,32,33], whereas, the present review describes the introduction to wearable devices capable of detecting different biomarkers in biological fluids aiming at the monitoring of human health. Lastly, future perspectives and challenges are also discussed while concluding the review.

## 2. Wearable Devices and Their Features

### 2.1. On The Basis of Their Design or Utility

The integration of wearable devices in personalized health services has gained significant attention since the beginning of the 21st century. Wearable devices can be classified as wearable bands (such as watches, gloves), wearable textiles (t-shirt, socks, shoes), wearables gears (glasses and helmets) and sensory devices for health monitoring [34,35,36,37,38,39,40] (Figure 2 and Figure 3). With the integrated miniaturized devices and advancements in technologies (microelectronics and wireless communication), wearable biochemical sensors have deeply embedded and become an integral part of our lives [41,42]. However, further development is the need in the future.

#### 2.1.1. Wrist-Mounted Wearable

As the name implies, Wrist-Wearable Devices (WWD) are usually worn on the wrist. WWDs for monitoring of physiological parameters have been developed, with the advantage of providing miniaturization and an improvement in battery longevity to convert raw signals into real-time interpretable data [41]. Recently, wrist-mounted wearables such as smartwatches or fitness bands have moved from basic accelerometer-based ones (like pedometer) to include biometric sensing. Commercially existing wrist-worn devices are either wristbands or smartwatches [43] and used as non-invasive human monitoring devices [38,43].

##### Wristbands

While there are similarities among wristbands and watches, wristbands are specifically designed to track human health and fitness activities and are popularly categorized as wrist-worn wearable devices [43]. In a typical design, a wristband do not have a display screen for notifications or limited features over the smartwatches which are aiming to replace the conventional watches. For example, Jawbone [44] made the UP4 band, which works on bio-impedance sensors to monitor activities such as walking and tracking with the ability to record the sleeping cycle. It also can capture signals such as heart rate, body temperature, and galvanic skin response (GSR) using various sensors (bio-impedance, triaxis) sensors located on the inner side of the band. However, UP4 does not have a screen display, and the data can be read via the smartphone app-enabled in a smartphone. Other than UP4, there are other bands such as Fitbit [45], and Huawei Talkband B3 [43]. On the basis of current market trends, the market for wristbands is expanding rapidly and there is an increasing interest in healthcare monitoring and well-being. In 2016, approximately 40 million device sales were predicted, which was after the smartwatches.

##### Wrist Watches

In modern life, smartwatches are one of the most important wearable device types. In 2016, Gartner reported [46] that the smartwatch sales in the wearables market were the second product over the smart devices collectively with 50 million units sold. Usually, a smartwatch monitors specific human physiological signals and biomechanics and therefore it acts as a fitness tracking device that helps users to log their daily activities such as automatically recording workout times, tracking heart rate, step counts, and calories burnt [47]. With the help of internal and external sensors integrated with a lithium-ion battery, smartwatches collect information and further transfer it to the cloud server or smartphone for analytics and readability.

The first commercially available non-invasive glucose monitor approved by the Food and Drug Administration (FDA) is owned by GlucoWathcfi biographer (Cygnus Inc., Redwood City, CA, USA) [48]. In this system, an electrochemical signal corresponding to the glucose concentration is extracted from skin interstitial fluid by reverse ionophoresis. Glennon et al. [49] introduced a smartwatch system, including fluid and storage systems to monitor sodium (Na^+^) content in sweat. Additionally, the device is capable of measuring daily activity including gestures, motion and patient monitoring.

Monitoring of high blood pressure (BP) or hypertension is one of the important and crucial adjustable risk factors to examine patient health status suffering from cardiovascular diseases (CVDs) [50]. Meanwhile, monitoring of arterial blood pressure (ABP) is promisingly an efficient way to monitor and control the prevalence of CVD patients. Therefore, monitoring of BP is one of the most important physiological parameters in the ambulatory setting and monitor an individual’s health status [51]. In conventional pulse wave sensors, a cuff-system is used to non-invasively monitor B.P. with optical, pressure, and electrocardiogram (ECG) sensors. These sensors encounter the limitations of large size, difficulty in handling, and inaccurate measurement in mobile position, which limits their wider utility [41].

To overcome the above constraints, Lee et al. [52] developed a wearable system with a Hall device that is able to detect the changes in the magnetic field of the permanent magnet and record the pulse wave data. This is a wrist wearable watch with the function of a pulsimeter without a cuff. Similarly, a skin-surface-coupled personal wearable device that captures waveforms of high-fidelity BP in real-time and communicates with a wireless system such as smartphones and laptops has been developed by Hsu et al. [53]. Ishikara et al. reported a photoelectron imaging (PPG)-based heart rate sensor, which detects changes in the heart rate and recognizes the possibility of overcoming motion artefacts in daily lives [54]. A smartwatch integrated with a gyroscope/accelerometer function can be useful to monitor and analyze tremor and balance dysfunction in Parkinson’s disease (PD) patients [55]. They assessed the capability of a smartwatch for the quantification of tremor in PD patients and evaluation of clinical correlation, its acceptance and reliability as a monitoring tool. Later, it was found to be promising.

Additionally, Tison et al. designed smart devices and developed an algorithm to detect atrial fibrillation (AF) from the data of the heart with the rate measured utilizing PPG sensor and accelerometer [56]. The wrist-worn wearables are the main contributors to making wearable products mainstream. The two main sub-categories of wrist-worn (wearable devices worn on the wrist) smartwatches and wristbands at the moment address two different user needs. The replacement of the traditional wrist-watches and their usage as an extension device for the smartphone vs. accurate and specialized tracking of a range of fitness activities with some overlap in basic fitness tracking functions. These two types of products are likely to merge in the future for day-to-day fitness tracking. Nonetheless, it is likely that more sophisticated fitness tracking wristbands will continue to exist for users who need advanced analytics.

##### Wrist Patches

A flexible and microfluidic-based patch system for real-time analysis of sweat samples was developed Nyein et al. [57]. This sensor is constructed on a flexible plastic substrate integrated with a special spiral-channel microfluidic embedded with ion-selective sensors. This system interfaces the sensing component and is capable of analyzing sweat with a printed circuit board (PCB) technology. The sensor could potentially monitor the concentration of ions (H^+^, Na^+^, K^+^, Cl^−^) and sweat rate, which further facilitates monitoring of human physiological and clinical conditions by sweat parameters. Moreover, there is still scope to improve the temporal resolution of the sensors, which could enable the ease and high-throughput in fabrication. Considering the requirement of soft and flexible WBs, which can imitate the skin surface, a wearable lab-on-patch platform constructed utilizing polydimethylsiloxane (PDMS) with an integrated microfluidic collection system was developed by Lee et al. [58]. In this design, antibodies specific to cortisol (MX210 Ab) were immobilized on a stretchable and conformable nanostructured surface with impedimetric-based detection. Under optimized antibody concentration level, the patch offers a detection limit of 1.0 pg mL^−1^ with a detection range up to 1 µg mL^−1^. The 3-D Au-nanostructure as a working electrode enables the higher sensitivity, even though the sensor has the limitation of Ag–Ab complex instability with no reproducibility [59]. To overcome the above instability concern, an artificial molecularly imprinted polymer (MIP) synthesized from copolymerization reaction for cortisol screening was reported by Parlak et al. [60]. The MIPs possess higher selectivity against cortisol as template, reversibility, robustness and reproducibility. The same group of researchers developed a device known as “SKINTRONICS” to determine the stress levels via electrodermal sensing of galvanic skin response [61]. This is a multilayer device with a wear time of 7 h with flexible hybrid skin-conformant features allowing the capture of real-time data. Currently, various skin-interfaced wearable-patch or sensing platforms are under the development stage, indicating a shifting focus toward flexible sensing [61].

#### 2.1.2. Head-Mounted Devices

Head-mounted tools are visual devices with hands-free capabilities, usually mounted to the user’s head [62]. This class of wearables has the highest number of research types like helmets, glasses, and caps. These devices are currently utilized in surgery, imaging, simulation; however, the commercial head-mounted wearables do not seem to be yet mature enough compared to wrist-worns [43,62]. In the literature, a number of head-mounted display devices are identified, which are currently utilized in surgery, imaging, simulation, education and as a navigation tool [62,63].

##### Eyeglasses

Smart glasses are wearable systems (WSs) that are a kind of head-mounted computer with a display property. For example, pulse-sensing smart glasses containing a photoplethysmography (PPG) sensor on the nose pad monitoring heart rate continuously were developed by Nicholas Constant et al. [64]. Further, to cross-validate the heart rate data, the pulse-glass sensor was compared with a laboratory ECG system during various physical activities by a participant. Eyeglasses integrated with a nose pad consisting of a lactate biosensor was developed to simultaneously monitor sweat lactate and potassium level in real-time [65]. These eyeglasses offer an inherent advantage of an interchangeable sensor. They have a large variety of nose-bridge amperometric and potentiometric sensor stickers. One example is that the lactate bridge-pad sensor is interchangeable with a glucose bridge-pad sensor in the monitoring of sweat glucose. These fully integrated wireless “Lab-on-a-Glass” multiplexed eyeglasses sensing platforms can be further expanded for the simultaneous monitoring of electrolytes and metabolites in sweat fluid. Integration of smart eyeglasses is additionally possible for the measurement of other human actions like barometers, accelerometers, gyroscopes, altimeters, and GPSs [41]. For example, Recon Jet, a sophisticated smart glass, is intended to capture health status while running or riding a bicycle by displaying information on the display. According to the literature, several smart eyeglasses have been designed for various applications such as tear biosensing [66] to detect vitamin and minerals, computational eyeglasses for sensing fatigue and drowsiness [67], medical use and health monitoring [68], EOG (electrooculography)-based human–wheelchair interface [69], sweat lactate biosensor using bienzymatic Gel-Membrane using eyeglasses [70].

##### Cavitas

Cavitas wearable sensors are attached to body cavities such as contact lenses and mouthguards. In Latin, “Cavitas” is the etymological origin of the word “cavity” [71]. These sensors provide information from the biological fluid within a body cavity. Numerous cavitas sensors have been reported for monitoring chemicals of biomolecules in tear fluid, and transcutaneous gases at eyelid mucosa. Additionally, the mouthguard sensors have also been investigated for real-time monitoring of chemicals in saliva. A mouthguard glucose sensor based on MEMS (microelectrochemical system) with enzyme membrane immobilized glucose oxidase was fabricated by Mitsubayashi et al. [72]. This sensor was capable of detecting glucose in artificial saliva over a range of 5–1000 µmol L^−1^ glucose with a stable and long-term real-time monitoring of more than 5 hr using a telemetry system.

Similarly, Kim et al. [15,73] demonstrated an enzyme-based biosensor integrated mouthguard for detecting salivary uric and lactate provided high selectivity and sensitivity. In neonates, monitoring of vital signs and symptoms, the development of portable and non-invasive health monitoring devices is of great interest as infants cannot speak about the discomfort or health complaints [74]. A pacifier biosensor operating as a wireless device for non-invasive chemical monitoring in the infant’s saliva was developed by Carmona et al. [75] to monitor glucose levels. Furthermore, saliva provides new potential for monitoring of metabolites in infants and neonates non-invasively.

##### Caps/Helmets

A group of Danish researchers designed a helmet that is used for the treatment of depression by reactivating body parts involved in depression and rapid recovery of patients by the transmission of weak electrical pulses to the brain [76]. Food and Drug Administration (FDA) has approved the helmet for depression treatment utilizing weak electrical impulses that are transmitted to brain part centric to depression [77]. The design of two heads-up display-based systems has with the ability to mitigate physiological conditions such as nausea, seizures, body posture have been also reported [78,79].

#### 2.1.3. E-Textiles/Smart Clothing

Smart clothing (E-Textiles) is derived from intelligent or smart materials with the ability to sense different environmental conditions and respond to stimuli such as thermal, chemical or mechanical changes. The first time, the concept “E-textiles” was defined in Japan in 1989 [80]. E-Textiles are an emerging interdisciplinary field of wearables with potential applications in healthcare, fitness, health and safety [81]. Worldwide, various material scientist groups are involved in developing conductive fabrics with embedded sensors on fabrics which is not the focus of the present review. These are fibers and filaments, consists of conductive devices and clothing material that is attached to or woven with the conductive tools, which can interact with the environment/human body [82]. Sensors provide a nervous system to detect signals, such as E-textiles incorporate sensors like electrodes sewing into fabric [82,83,84] and these integrated biosensors are used to analyze biofluids [84]. E-textiles combine a high level of intelligence and further divided into three types [82], i.e.,
(a)Passive E-textiles: Which can sense the environment/user based on sensors integrated(b)Active E-textiles: Reactive nature and it can sense external stimuli from the environment, integrated with an actuator function and a sensing device(c)Very smart E-textiles: Ability to sense, react and change under given circumstances

Usually, E-textiles possess three components; a sensor, an actuator and a controlling unit [41] and are involved in the monitoring of human physiological signals, biomechanics and physical activities [43,85]. Liu et al. [84] fabricated an enzyme-based detection system to detect glucose and lactate by integrating the glucose and lactate-oxidase enzyme coupled electrodes into the fabric. Additionally, the same group of researchers [85] also developed a living material and a glove, which is integrated on the hydrogel-elastomer hybrids with genetically engineered bacteria including genetic circuits to provide a desirable function to the material. Herein, the chemically-induced various bacterial cell strains were encapsulated in a hydrogel chamber. Bacterial strain and environment interact via a diffusion process. An inducer (IPTG, Rham) contact with the bacterial sensor programmed on fluorescence response is activated. The biosensor constructed utilizing synthetic biology has promising potential in healthcare and the environment due to its mechanical flexibility and low cost.

E-textiles are also used to monitor physiological signals such as heart rate (HR), temperature and breathing rate [86,87,88]. Mishra et al. [2] designed a wearable gloved-based electrochemical biosensor on the stretchable printable enzyme-based electrode, which can detect the organophosphate (OP) nerve-agent compounds. Stress-enduring inks are used for printing the electrode system. A long serpentine connection was used in order to connect the wireless electronic surface. Glove design consists of a typical three-electrode system with a carbon-based counter electrode at an index finger, working electrodes, reference Ag/AgCl-based electrode, and a thumb-printed carbon pad. Herein, the index finger contains an organophosphorus hydrolase layer and acts as a sensing finger and the thumb is a sample collector/sampling finger. Later, the practical utility of lab-on-a-glove was demonstrated in defense and food security applications. Villar et al. [89] developed a hexoskin wearable vest, which is capable of monitoring HR and BR during daily activity. On the other side, to measure the walking ability and gesture, an electronic shoe has been developed to measure lateral plantar pressure, heel strike and toe pressure, and this helps to record essential information to distinguish among gait phases [90]. A conductive-textile-based wearable biosensor for BR sensing based on the capacitive sensing approach was developed by Kundu et al. [91]. A t-shirt is usually worn at the abdomen or chest position, where the respiration cycle is measured by the capacitance of two electrodes placed on the inner anterior and posterior sides of a T-shirt. Hyland et al. [92] reported the generation of wearable thermoelectric generators (TEG) for human body heat harvesting. In a typical design, TEG was used to harvest electrical energy from human body heat that further power wearable electronics.

A smart shirt-based biosensor was designed to measure electrocardiogram (ECG) and acceleration signals for continuous and real-time health monitoring [93]. In a typical design of the shirt, it consists of a sensor for real-time monitoring of the health data and a conductive fabric as electrodes to obtain the body signal. These wearable sensors are designed in a way so that they fit well into the shirt with small size and low power consumption to reduce the battery size. To cancel the artefact noise from the electrodes made up of conducting fibers, an adaptive filtering method in the designed shirt was designed and tested to get a clear electrocardiogram signal while running or performing physical exercise.

#### 2.1.4. Chest-Mounted Devices

The monitoring of falling and postural disability of individuals are primary concerns for caregivers or health workers [94]. Two alert systems, i.e., the Life Alert Classic by Life Alert Emergency Response Inc [95] and the AlterOne medical alert system [96] are commercially available for safety monitoring. In these devices, a pendant is integrated with a push button and pressing the button transfers the message wirelessly to a remote location. Parallelly, the Wellcore system employs advanced microprocessors and accleratormeter units for monitoring the postural movement [97]. This device is able to differentiate between normal and falls body movements and communicate further with a remote center. MyHalo^TM^ by Halo Monitroing^TM^ is also a chest-worn device used for monitoring the heart rate, sleep pattern and temperature, etc. [40,98]. In conclusion, a device with an integrated system on a mobile device that is equipped with a balance sensor that triggers automatic dialing on emergency contact in case of falling will be useful in an impactful manner.

### 2.2. Bio-Multifunctional Smart Wearable Sensors (WSs)

In the construction of WSs, the selection of nanomaterials with mechanical compatibility is one of the essential factors to mimic the biofunctions. Monitoring of various biological signals including physical, electrophysiological and gait abilities as the vital sign of existing fatal diseases are key factors. Over the last few decades, the readiness of wearable health monitoring devices has permitted the early diagnosis of these vital biological signals [99]. The availability of an ideal material can improve the performance and wear resistance of the wearable sensor and enhancing its utility. This section will discuss the types of bio-functional materials used in wearable sensors.

#### 2.2.1. Self-Healing Flexible Wearable Sensors

Currently, wearable medical devices are limited by their robustness due to the ease of damage of biosensor components, which alter the function and further reduce their performance, shelf-life and electronic properties [100]. For an ideal bio-multifunctional wearable biosensor being an intelligent sensor, they not only retain their electronic functions but also possess self-repair properties to maintain their internal physical characteristics upon minor micromechanical damage [101]. Wearable electronic devices used on the skin must include the characteristics of self-repair without any external stimulation (e.g., heat) to restore their mechanical and electrical connections [102,103,104]. Several self-healing flexible sensors based on conductors and polymers have been investigated [101,105]. Despite rapid development in the field of self-healing polymeric materials, only a few of them have been used in the field of flexible wearable electronics [33]. A variety of composite materials, which are filled with conductive particles or healing agent-loaded capsules, are used to achieve self-healing ability.

He and co-workers reported the development of self-healing electronic sensor-based prepared by incorporating ionic liquids into self-healing polymer channels [106]. In this design, the leakage of ionic liquids at a breaking state is avoided due to the capillary effect. Bandodkar et al. [107] reported synthesis and incorporation of a conductive ink containing carbon (45%) and an acrylic varnish binder (5%) into self-healing electrochemical and wearable biosensors. Bao and co-workers designed a self-healing conductive composite for self-healing medical devices [108,109]. It described a rubber-like conductive composite composed of inorganic micronickel (μNi) and organic supramolecular polymeric particle that possesses an electrical and mechanical self-healing mechanism driven via the recombination of hydrogen bonds between cut surfaces. Jiang et al. [110] developed a flexible sandwich structural strain sensor. This sensor is fabricated by sandwiching a layer of silver nanowires (AgNWs) decorated polymer with self-healing properties into layers of PDMS (polydimethylsiloxane). This design provides good stability and stretchability. The fractural tensile stress of the self-healing polymer increased to 10.3 MPa with the elongation at a break of 8%. Above all, several reports have been already published which potentially explain the advancement in the materials or nanocomposite used in wearable biosensors [111,112].

Recently, hydrogels [113,114] have gained promising attention in advanced wearable sensors due to their mechanical properties. However, manufacturing a skin-like stretchable and conductive hydrogel with desired synergistic characteristics of stretchability, higher self-healing capacity and an excellent sensing performance still remain a challenge [115]. Chen et al. [33] developed a 3D network of electro-conductive hydrogel through a two-step process and its application for human motion detections. In this work, the cellulose nanofibrils oxidized by 2,2,6,6-tetrametylpiperidine-1-oxyl (TEMPO) were homogeneously dispersed into polyacrylic acid (PAA) hydrogel with ferric (Fe^3+^) ions as a crosslinker to synthesize the TEMPO-oxidized cellulose nanofibrils/polyacrylic acid hydrogel. Later, a polypyrrole conductive network was incorporated into the synthetic hydrogel which forms a polymeric 3D-network and is interlocked by strong hydrogen bonds and ionic interactions. This further improves the mechanical stability, texture, self-healing ability with electrical and mechanical healing efficiencies of ~99.4% and 98.3%, respectively. Zhang et al. [116] developed GO-based hydrogel prepared from GO, polyvinyl alcohol (PVA) and polydopamine (PDA) with improved mechanical and electrical properties. These hydrogels were assembled into wearable sensors employed for real-time detection of human motions (large and small scale motions). This was achieved through recombination and fracturing of the rGO electrical pathway.

Despite the important features of hydrogels, the fragility and brittleness of the hydrogels are two important obstacles in their further applications in wearable devices. These problems can be overcome by strategies like double and interpenetrating networks such as double hydrogels, nanocomposite (NC)-based and double crosslinked hydrogels with strong mechanical properties and stability in extreme conditions [117].

Recently, the introduction of dynamic polymer materials with self-healing capacity based on the reversible bonds and dynamic interactions gained significant attention [33]. Cao et al. [118] reported the development of a bio-inspired skin-like material that is transparent, conductive, and self-healing under dry and humid conditions. These polymeric materials are combined with chemically compatible ions providing a gelatinous character with aqueous, stretchable, self-healing electronic skin nature. Very interestingly, this material possesses self-healing ability under various water environments (i.e., deionized, seawater, extremely acidic and alkaline solutions). Other researchers have also reported the formation of reversible nature dynamic materials and their utility in wearable electronic devices [119,120,121,122].

#### 2.2.2. Biocompatible Wearable Sensor

Wearable biosensors are directly exposed to the human body; therefore, it is expected not to pose any kind of additional health risk to human life. Henceforth, it is essential for the wearable biosensor to be biocompatible to avoid the occurrence of an immune response [123], which makes the biocompatible material as preferable materials for smart wearable sensors. Recently, bioresorbable silicon-based multifunctionality electronic sensors for the brain have been proposed by Kang et al. [124]. They confirmed that no sign of glial cell response was found for 8 weeks after implantation, which is an indicator of material biocompatibility. Additionally, the construction of a biocompatible conductive polymer-based implantable pressure-strain sensor was reported by Baoutry et al. [125]. This sensor is able to measure pressure and strain individually via two vertically isolated devices and identify the pressure due to a salt (12 Pa) and a strain of 0.4% without interference. In vivo studies of the device exhibited good functionality and biocompatibility in rat models, this indicates the potential applicability of the method in real-time monitoring. Later, the same group of researchers also proposed the design of a pressure sensor made-up of biodegradable materials constructed on fringe-field capacitor technology to measure the arterial blood flow in both contact and non-contact modes [126]. Operation of the biosensor was described by a custom-made in vivo artificial artery model and advantageous in real-time post-operative blood flow.

Nowadays, designing and integrating the nanoscale material in health and medical-related issues is of prime importance [127]. However, the interaction between a single substance and the human biological system is quite challenging to envisage due to a series of specific biological reactions inside the human body. Precise design and in vivo measurements are very critical to confirm and assure the biocompatibility of material in a particular application [124]. To overcome, an excellent choice to select natural biocompatible polymer or materials such as cellulose [128], chitin [129], alginate [130], polydimethylsiloxane (PDMS) [131,132], plolyurethane (PU) [133] as they are non-toxic in nature. In recent years, various biosensors utilizing these biomaterials have been widely reported [134]. Chitosan is one of the highly notified examples with good tensile and conductive properties [135]. A simplified strategy to design multifunctional biomaterials integrating with natural chitosan and graphene was developed by Ding et al. [136]. Biocompatible composite materials possess several advantages over other structural materials such as higher sensitivity, rapid response and achieving as much as a low limit of detection (20 ppb), and their applicability to design chemical sensors for real-time diabetes monitoring.

#### 2.2.3. Biodegradable Flexible Sensors

According to recent reports, biodegradable devices are promising devices that enabled the advanced level of health monitoring and reduction in the generation of electronic waste [137,138,139]. These WB-based technologies have resulted in the decrease of ill effects on human health.

Boutry et al. (2015) designed a sensitive and high-performance wearable pressure device using biodegradable conducting polymers for cardiovascular monitoring [140]. Higher sensitivity with faster response time allows integrating the designed sensor for continuous cardio-vascular monitoring such as a recording of blood pulse signals. It may also be used in biomedical applications to avoid surgical interventions. Approaches to design and construct flexible biosensors primarily require a strategic approach to tune materials for biomedical application. Pal et al. (2016) reported an easy method to fabricate poly(3,4-ethylenedioxythiophene): poly(styrene sulfonate) sensors on a fully biodegradable and flexible silk protein fibroin support utilizing photolithography [141]. Due to the conductivity of micropatterns as the working electrode, the biosensor showed excellent electrochemical activity and stability over a few days in the detection of dopamine and ascorbic acid with higher sensitivity. However, these sensors are liable to be attacked by an enzymatic reaction. Rogers and co-workers are pioneered in the research of implantable silicon-based transient electronic devices which are used in wearable devices [142]. They design and constructed a silicon-based sensor with silk as a substrate for antibiotic studies. Due to its lightweight, inexpensive, eco-friendly and flexible nature, the use of paper has gained significant attention for the fabrication of flexible and wearable sensors [143]. Jung et al. (2015) reported the construction of a high-performance wearable sensor utilizing cellulose nanofibers. Additionally, Zhang et al. (2012) reported the use of rice-paper as a separator device fabrication [144], which emphasizes the integration of rice paper membrane in green electronics due to low cost, good flexibility, low resistance and porous structure.

### 2.3. Microfluidic-Integrated Wearable Biosensor

#### 2.3.1. Colorimetric-Based Wearable Sensors

Ideally, the real-time sensing platform must be simple, easy to use, fast and economical. The appearance of change in color upon chemical reaction between analyte and recognition site offers rapid detection of target molecule such as in colourimetric-based sensing systems [145,146]. Utility of colorimetric biosensors for detection of ions (H^+^, Na^+^, K^+^, Ca^2+^, Cl^−^) [147], single-molecule [148,149], microbes [150,151], proteins [152] has already been documented. In recent time, the colorimetric detection utilizing standard spectrophotometry tools and integrated high-definition cameras of smartphones can also be performed [152,153]. Furthermore, this system may also be mixed with textile materials to use as textile-based fluid handling platforms collecting and analyzing the sample for real-time analysis, which removes the need for mechanical micropumps. In this regard, the first example of a microfluidic-based colorimetric wearable biosensor integrated on polyester/lycra textile was reported by Morris et al. [154]. It has fluid transport characteristics and is influenced by the density and the ratio of the two materials. On fabric, the pH sensor was constructed via functionalized fabric microfluidic with pH dyes sensors with great fluid-transport characteristics. Curto et al. (2012) used an alternative approach, wherein, the microfluidic system was integrated on a cotton thread to further facilitate the sweat transport to measure the real-time change in pH of sweat sample [155] and equipped with an LED-based detection system. To measure sweat pH, a cotton-based textile modified with organic silicate was demonstrated by Caldara et al. [156].

Introducing the paper for the construction of wearable biosensors have been thoroughly investigated, providing the advantage of lightweight, higher wicking, and ease of functionalization [157]. Mu et al. (2015) proposed a paper–skin patch-based screening method to detect sweat anions such as lactate, chloride, and bicarbonates in patients suffering from cystic fibrosis [158]. These papers can be incorporated into the adhesive skin patch to provide seamless skin contact. Additionally, the paper-based devices also found interest in the detection of pH levels in saliva and sweat samples to estimate and monitor the water/dehydration level and prevent decalcification of enamel, respectively [153]. Considering the potential of paper, various colorimetric wearable devices constructed using plastic-based microfluidic have been reported. Matzeu et al. (2015) reported the development of a colorimetric detection method to study the sweat rate while doing exercise [159]. On video mode, an estimation of sweat flow rate in several cycles showed the effectiveness of the method. A PMMA-based microfluidic wearable biosensor featured the collection of sweat to direct towards an active area where pH can be measured proposed by Curto et al. [155]. Herein, the authors used four different ionogels (ionic liquid polymer gels) functionalized with individually different pH dyes with different pKa covering the full spectrum of sweat pH.

Furthermore, the development of smartphone-based wearable applications explained the simple and direct way for real-time estimation of sweat rate. Recently Xiao et al. (2019) reported a microfluidic chip-based wearable biosensor for monitoring the sweat glucose level [160]. In this device, five microfluidic channels connected to the detection microchambers, which routed the excreted sweat from the epidermis to microchambers and having a check valve to stop the backflow of reagents from microchambers. The glucose sensing was based on the reaction with pre-embedded GOx-peroxidase-o-dianisidine and colour change due to enzymatic oxidation correlating with the glucose concentration. This system can detect glucose from 0.10 to 0.50 mM with LOD of 0.03 mM. A smartphone integrated image capturing wearable microfluidic device for the detection of glucose, lactate, chloride ion in sweat was reported by Koh et al. [161]. The developed colorimetric device showed similar results compared to conventional analyzes. According to stress analysis, the sweat patches remained intact even if used under outdoor physical exercise. Interestingly, the PDMS-based microfluidic wearable systems have been proven as effective platforms in collecting the sample and analyzing them. Based on that, Choi et al. (2017) developed a thin, soft wearable microfluidic device with the ability to mounts onto the surface of the skin that enables the collection of sweat in micro reservoirs via microchannels with integrating valves open at various pressures [162].

#### 2.3.2. Electrochemical-Based Wearable Sensors

An emerging electrochemical-based biosensing platform represents an alternative approach to colorimetric sensors inherent with the characteristics of the higher sensitivity and selectivity for a large number of metabolites [163,164,165]. Advancement in nanotechnology, polymer science, and integration of inorganic materials has further provided an improved sensitivity and limit of detection of electrochemical biosensors [155]. In the absence of microfluidic, integrating an electrochemical biosensing platform with pilocarpine iontophoresis mechanism enables an alternative route for sweat analysis. Alternatively, these techniques are not suitable for sweat analysis at the resting time due to the absence of sweat secretion.

The first example of wearable electrochemical biosensors for sweat analysis in real-time was proposed by Schazmann et al. [166]. This sensor was integrated with an ion-selective electrode for sodium-ion and a fabric-based pumping system for collection and directed the sweat towards an active area. In this device, the glass electrode was used, which is the most suitable material for wearable application. Matzeu et al. (2016) demonstrated a screen-printed electrode integrated with PMMA and adhesive to design a compact and flexible device for sodium analysis in sweat [167]. This electrodes system was further incorporated into a microfluidic system with a read-out system for continuous real-time monitoring. A T3 incorporating cryogels and screen-printed electrodes were used for monitoring of ethanol [168]. Detection strategy utilized the oxidation of substrate by alcohol-oxidase enzymatic reaction for an amperometric detection for alcohol determination. Martin et al. (2017) reported the development of a skin-mounted wearable device that integrates with a flexible microfluidic and electrochemical detection mechanism for detecting glucose and lactate in sweat samples [169]. Even though various advancements have been reported, the commercial aspects of wearable devices are still under consideration and there is more to be explored.

## 3. Application to Detect Biomarkers in Biofluids

Since a few decades ago, due to the inherent characteristics and potential utility of wearable biosensors and biomedical devices, these devices have proven their utility in the detection of biomarkers, drug metabolites and hormones in various biological fluids and matrices (Figure 4).

### 3.1. Saliva-Based Wearable Biosensors

Since the last few decades, an interest in saliva as a diagnostic fluid has gained tremendous attention due to the presence of various disease-signalling biomarkers that reflect the health status of humans [171]. The existence of various disease-signalling salivary biomarkers that accurately reflect normal and disease states in humans and therefore the sampling benefits compared to blood sampling are a number of the explanations for this recognition [17,171,172]. Various biological markers in saliva diffuse from the bloodstream via transcellular/paracellular paths, making saliva a mirror of human health. These biosensors offer an alternative pathway to the blood analysis for monitoring of human metabolites such as hormones and proteins [173]. Saliva is a highly complex biofluid with high protein content produced by the parotid gland composed of some important constituents such as drug metabolites, enzymes, microbial flora, and hormones [173,174,175]. Previously, these biomarkers have been used in diagnostics; however, there are few reports on wearable saliva biosensors likely due to the biofouling of rich salivary protein content and low concentration of the analyte to be detected. Notwithstanding, wearable mouth biosensing platforms can offer an attractive and painless route for obtaining dynamic chemical information from saliva. Wearable devices for oral use require the integration of biosensors and an electronic interface into an orally mounted device, for example, a mouthguard or denture-based system [10].

In the late 1960s, the first example of salivary sensors was proposed by Graf and Mühlemann [176] to monitor pH and fluoride ion activity on tooth enamel. These platforms were subjected to the risk of leakage of internal sensor solutions. Oral biosensing is pioneered by Mannoor et al. (2012), who reported the generation of graphene-based nanosensors designed on printed silk and used for passive detection of bacteria [177]. With the recent progress in salivary diagnostics, wearable salivary biosensing has emerged out as a potential strategy [178]. For the first time, Kim et al. (2014) reported the development of an electrochemical biosensor based on integrated screen-printed enzymatic electrodes on a mouthguard for monitoring salivary lactate [73]. This biosensor is highly selective to detect salivary lactate electrochemically using lactate oxidase (LOx) enzyme immobilized of screen-printed surface used for low potential detection of peroxide product. The researcher carefully protects the sensor against biofouling and confirmed against undiluted salivary samples by electropolymerized o-phenylenediamine for non-invasive and continuous monitoring of individual health. The same group of researchers (Kim et al., 2015) further continued with the development of a non-invasive mouthguard-based uric acid oral-cavity biosensor incorporated with miniaturized electronics featuring a potentiostat, a microcontroller and a Bluetooth low energy (BLE) transceiver [15]. In this system, a transducing element is constructed via Prussian blue (PB) embedded in a carbon electrode on PET and electropolymerized OPD cross-linked with uricase act as a biorecognition site, which allows non-invasive monitoring of salivary uric acid similar to blood uric acid (a common biomarker for hyperuricemia, gout and renal syndrome). These biosensors exhibited various advantages such as wearability, ease of operation, and renewability.

Arakawa et al. (2016) reported the development of a detachable “cavitas sensor” for monitoring salivary glucose intergrade on a mouthguard platform [72]. The biosensor was constructed on a glucose oxidase (GOx) modified poly(ethylene terephthalate) glycol surface integrated with a wireless transmitting system. Interestingly, this device enables a telemetric-based measurement of salivary glucose in an artificial salivary system in the range of 5–1000 µM. The developed system is highly stable and provides real-time monitoring for more than 5 hrs with a telemetry system. Establishing the correlation between blood and salivary glucose reflects diffusion and active transport of blood components to the salivary gland [179], which offers a highly efficient route for glucose monitoring, mainly for the patient suffering from hormonal and neural balance and disorder like diabetes. A correlation of R^2^ = 0.64 in healthy and R^2^ = 0.95 in diabetic individuals was calculated by Soni et al. [180]. Another example of a wearable device based on an oral-cavity-based platform was recently demonstrated for in-mouth operation by Tseng et al. [181]. The sensor was constructed utilizing porous silk and hydrogel capable of wireless monitoring of food consumption and oral cavity for alcohol content, pH, sugars, salinity, etc. Monitoring of ions or salts in the biological system is essential. Considering that, a sensing system for in vivo oral monitoring was developed. A user-comfortable system using ultrathin stretchable electronics for sodium-ion monitoring via long-range telemetric system [182]. Despite several reports available on the development of oral-cavity based wearable biosensors [183,184], a critical evaluation is mandatory, ensuring the safety and reliability of the developed system. Some of the challenges are analyte contamination by food, salivation rate, inappropriate correlation of analytes, overcoming these challenges surely will improve the practical utility of saliva-based biosensor for monitoring of potential biomarkers present herein.

### 3.2. Tear-Based Wearable Biosensors

Similar to saliva and sweat, human tears are an important and complex biological fluid, which is composed of various proteins, electrolytes, metabolites and more than 98% of water [185]. Multiple components of tears are potentially useful to diagnose human metabolites. Since the 20th century, the development of tear-based devices has gained prominent attention; however, this field still has more potential to explore in wearable devices for tear monitoring. Contact lenses are an appropriate system to collect tears without any damage to the eye and are in direct contact with the basal tears [185,186]. They can be easily integrated with the necessary biosensing systems. Initially, the contact lens-based sensing platforms were developed for glucose monitoring in tear based on the interaction of glucose with concanavalin A (or phenylboronic acid) derivatives via optical measurements [187,188].

A first successful example of a contact lens-based wearable sensor was reported by Shum et al. (2009) [189]. In that study, a microfabrication technique for constructing on-body testing contact lens-based sensor with amperometric principle was used. The sensor used a polymeric substance, i.e., indium tin oxide (ITO) for immobilization of GOx, platinum working and counter electrodes with Ag/AgCl as reference. For practical application, the utility of the sensor was validated for hydrogen peroxide and glucose monitoring in the range of 10–20 µM and 0.125–20 µM, respectively. Similar to this approach, an enzymatic sensor was developed for lactate monitoring in tear fluid on a polymer substrate molded into a contact lens shape [190]. Under this strategy, LOx was immobilized on platinum sensory structures utilizing crosslinking chemistry with glutaraldehyde and bovine serum albumin. The developed sensor showed a quick response time of 35 s with an average sensitivity of ∼53 μA mM^−1^ cm^−2,^ and the response is stale up to 24 h.

Mitsubayashi et al. (2011) developed a contact lens-based biosensor for in situ monitoring of glucose in tear fluid [191]. In this design, silver (Ag) and platinum (Pt) metals were sputtered onto a polydimethylsiloxane (PDMS) substrate integrated with counter (Pt) and reference (Ag/AgCl) electrodes. The electrodes were fixed on the surface of contact-lens using PDMS. A copolymeric mixture of 2-methacryloyloxyethylphosphorylcholine and 2-ethylhexylmethacrylate (PMEH) was used for the immobilization of GOx. The CL biosensor exhibited a promising relationship between the output current signal and glucose concentration ranging from 0.03 to 5.0 mM with a correlation coefficient (R^2^) of 0.999 and also successfully employed utilizing a rabbit model. Very recently, a contact-lens-based sensing platform to detect lysozymes in tear-fluid samples was reported by Ballard et al. [192]. In this report, commercial contact-lenses were used as a sample collector. This study was conducted over a group of nine healthy human participants for a period of two-weeks. Lysozyme concentration was found to correspond to the increasing fluorescent signal captured using a time-lapse imaging system. They observed an increase in lysozyme concentration from 6.89 ± 2.02 μg mL^−1^ to 10.72 ± 3.22 μg mL^−1^ on inducing digital eye strain while playing mobile games. They were comparing that a lower lysozyme concentration, i.e., 2.43 ± 1.66 μg mL^−1^ was reported in a patient with dry eye disease. Later, the system was found to be non-invasive, easy-to-use, economical.

Recent advancements in wearable contact lens biosensors indicate the use of smartphones for the detection of the analyte of interest in tear fluids [10,193]. Elsherif et al. (2018) designed a hydrogel-based sensor with a photonic microstructure, which was attached on top of a commercial contact lens [194]. The reflective power of the lens was recorded with a smartphone corresponds to the change in tear glucose level. This design could offer fast and sensitive detection of glucose with ease of fabrication. These platforms could provide an attractive alternate platform to electrochemistry-based contact lens biosensors, which help in the miniaturization and readability of the device. In addition to contact lens-based biosensors, an electrochemical sensor constructed from multiple electrodes coated with a layer of protective polysaccharide hydrogel matrix was reported by NovioSense [194]. This sensor is placed at the inferior conjunctival fornix, allowing continuous access to the tear fluid, which helps in continuous glucose monitoring coupled with wireless transmission and no discomfort to the patient.

Overall, the tear-based biosensor is primarily focused on the monitoring of glucose; however, there is significant potential for the non-invasive detection of some other physiologically important biomarkers. Extending the application for other analytes, the analyte whose concentration in tear fluid is in close resemblance with the blood can be incorporated. On the other side, the finding of suitable power supply and their appropriate size is also a technical challenge in tear-based wearable biosensors. To overcome this, biofuel cells (BFCs) is considered as an effective way to generate power in situ [195]. Other than glucose, ascorbate, lysozymes, and pyruvate are the best examples of biofuel and analyte to be detected. Falk et al. (2012) studied tear-based BFC [196]. Herein, the researchers proposed a successful design of BFC constructed on nanostructured microelectrodes coated with gold nanowires and tested on the human tear. However, this method was unsuccessful in establishing the correlation with blood concentrations, sampling size and other side effects.

### 3.3. Sweat-Based Wearable Biosensors

Sweat glands are entirely distributed across the human body and are one among the important biofluid human sweat provides potential information of patient health status, which may be utilized in non-invasive and wearable sensing [197]. Additionally, the presence of varied metabolites, electrolytes (Na^+^, K^+^, NH^4+^, Ca^2+^), hormones, environmental contaminants present on the skin and its physiology provides the most viable sites for sampling and monitoring the metabolic diseases [198,199]. These wearables sensors are quite useful and possess the power for real-time health monitoring. The presence of biomarkers in sweat might be of considerable interest for non-invasive individual health monitoring, for example, hydration level, disease status (like diabetes, cystic fibrosis). In-situ non-invasive sweat monitoring at the epidermis layer further eliminates the priority associated with sampling in blood without damage to the skin layer. However, additional validation is additionally required to verify the clinical value of sweat as a diagnostic biofluid [10,198,200].

Subsequently, epidermal-based sensing has targeted the determination of an interested analyte in ISF fluid as beneath skin cells are surrounded by ISF, where a correlation might be established between ISF concentration of analyte and blood that diffuses directly from the capillary endothelium [201,202]. Monitoring of analytes in ISF required analytes to be present on the skin, which will be accomplished via reverse iontophoresis. Whereas, the extraction mechanism influences the accuracy of the method. With a cavernous understanding of sweat chemistry and metabolite transport mechanism, advancements in sweat sampling and detection technologies could accelerate the sensible utility of sweat-based diagnostic opportunities. Broadly, sweat based biosensors are divided into two groups, i.e., textile/plastic and tattoo (epidermal)-based systems [203] and counterparts with their advantage and disadvantages.

Epidermal wearables devices offer better skin contact; moreover, they possess a shorter life-span over textile-based biosensors. The first time, the epidermal-based biosensors were developed by Roger’s group to continuously monitor some physical parameters [204]. Windmiller et al. (2012) adopted and combined the strategy with biorecognition mechanism to demonstrate the development of a “temporary transfer tattoo (T3)”-based electrochemical sensors being an electronic skin to physiologically monitor chemical constituents [205]. Herein, Wang and co-workers reported the primary example of real-time non-invasive lactate monitoring employing a flexible printed tattoo (T3)-based electrochemical biosensor which resembles an electronic skin or skin-worn biosensor [206] as shown in Figure 5. This biosensor could detect the lactate up to 20 mM possess strength against mechanical deformation expected from epidermal wear. This device was successfully demonstrated for lactate monitoring within the sweat sample of human subjects during prolonged cycling events and reflects the change in lactate production during exercise. These wearable devices could provide useful insight into individual physiology. Based on the same concept, Wang and co-workers (2016) demonstrated the development of a tattoo-based wearable sensor (Figure 6) for on-invasive alcohol monitoring in induced sweat samples [168]. In this design, the skin-worn platform was integrated with the iontophoretic system using flexible wireless electronics. Detection of alcohol was based on the transdermal release of pilocarpine inducing sweat via iontophoresis and amperometric detection of ethanol via alcohol-oxidase enzyme with Prussian blue. For practical application, this skin-worn sensor on the human body exhibited significant differences within the current response with and without alcohol consumption, reflecting an increase in the ethanol levels.

Bandodkar et al. (2012) developed a completely unique tattoo-based solid-contact ion-selective electrodes (ISEs) for non-invasive monitoring of pH with potentiometric principle [199]. Herein, a combination of two different techniques, i.e., screen printing and solid-contact polymer ISE methodology were used to design a temporary transfer tattoo paper. The resulting sensor showed a rapid response and better sensitivity against a good range of pH changes with a 4.72% RSD and no carry-over effects. From the same group of researchers, Mishra et al. (2018) [207] reported the development of the first potentiometric tattoo biosensor for monitoring G-type nerve agents in real-time. This biosensor offers a faster response and better selectivity towards the detection of diisopropyl fluorophosphate (DFP) in both the liquid and vapor phases. DFP may be a fluorine-containing organophosphate (OPs) structurally similar to chemical warfare agents, i.e., sarin and soman. For detection, the biosensing system relies on the pH-sensitive polyaniline (PANi) coating over a flexible printed transducer used for the detection of a proton released from enzymatic hydrolysis of DFP by organophosphate hydrolase (OPH). This sensor showed a promising dynamic range and higher selectivity against DFP and therefore the design of wearable biosensor reflects a substantial approach for on-body detection of G-nerve agents.

Furthermore, the tattoo ISE sensors endure repetitive mechanical deformation, which is a key requirement of wearable and epidermal sensors. The flexible and conformal nature of the tattoo sensors enables them to be mounted on nearly any exposed skin surface for real-time pH monitoring of the human perspiration, as illustrated from the response during strenuous physical activity. The resulting tattoo-based ISE sensors offer considerable promise as wearable potentiometric sensors suitable for diverse applications.

Rose et al. (2014) reported the development of a wirelessly powered patch-type wearable sensor [208] based on adhesive radio-frequency identification (RFID) (Figure 7). This sensor can mimic the human skin and demonstrate for biomarkers monitoring within the sweat sample. In this design, an electronic circuit together with an RFID was fabricated on an adhesive patch and employed for the potentiometric measurements of analytes in sweat samples and results can be read out on Android smartphone application wirelessly (Figure 7a,b). The response is often analyzed on an android smartphone app with an accuracy of 96% at 50 mM Na^+^ concentration. Later, the same group of researchers introduced an epidermal patch-based amperometric platform constructed on the screen-printed electrode (SPCE) (Figure 7c). This epidermal patch offers simultaneous health monitoring and L-lactate detection in sweat sample introducing Prussian blue (PB) [19]. A well-planned extremity of the design was the integration of all components (sensory part, microcontroller, a wireless communication module, potentiostat) on a single chip. Karyakin et al. [209] developed a nonenzymatic impedimetric sensor on SPCEs electropolymerization for 3-aminophenylboronic acid with lactate as a template. Moreover, this was not a perfect design of WBs but could detect lactate between 3.0 and 100 mM with a LOD of 1.50 mM with a response time of 2–3 min and a shelf-life of 6 months. Figure 7d represents lactate profiling of 17 lactate samples compared with enzyme-based lactate sensors.

Additionally, several metabolites present in sweat can serve as biofuels by transferring power to the wearable biosensor. Recently, Wang and co-workers published a report on sweat-based BFCs with a T3 technique and transferred to human skin [210]. Herein, LOx immobilized CNTs used as bioanode and Pt modified carbon electrode as biocathode with O_2_ as substrate. Based on the lactate concentration during perspiration, the constructed biosensor exhibits a high-power density of 5–70 W/cm^2^.

### 3.4. Implantable and Subcutaneous Wearable Biosensors

Advancements in the field of developing devices for subcutaneous wearable monitoring of intercellular fluids (ICF) have gained significant attention in recent years [83,211,212]. For example, various devices are available for subcutaneous monitoring of glucose concentration in diabetic patients [213]. Usually, ISF surrounds the cells beneath the epidermal layer and helps in the regulation of optimal organ homeostasis. Various important components of ISF such as ions (Na^+^, K^+^, Cl^−^), metabolites (lactate, glucose, hormones) are the preferable target for the construction of miniaturized wearable devices for real-time monitoring of ISF mainly the components present in it [214]. Integration of microneedles and micro-dialysis probes technologies brought lots of attention in the field [215].

A typical design of micro-dialysis, it uses a coaxial microfluidic probe made-up of polymeric material having a desired cut-off pore size, which established direct contact with the tissue. The analyte concentration gradients across the membrane and tissue are further responsible for analyte diffusion through the membrane inside the probe. To determine the analyte concentration from the dialysate, either a specified volume of dialysate is collected for downstream analysis for quantifying the analyte concentration or real-time online methods such as HPLC, LC-MS are used [17]. To date, the micro-dialysis techniques are limited by factors of real-time information or the need for bulky analytical tools for analysis [216]. Gowers and co-workers reported alternate to the above-mentioned limitation [217,218]. It was simple to achieve delayed high temporal analysis via tubing instead of vials used for storing dialysate, which made it easy to continuously measure lactate and glucose by an implanted probe with an estimated time resolution of 30 s.

Moreover, this approach deals with the demerit of delay in the collection of analysis. On the other side, the microneedle technology is based on the use of an array of micro-sized needles fabricated from rigid materials such as silicon, biocompatible polymers or hard plastics [219]. Integration of microneedles technology allowed a simultaneous use for analyte detection in ISFs and in-situ drug delivery [220]. For biomarker monitoring and performing function of targeted drug delivery, microneedles-based approaches have been used in wearable patterns [221,222].

A practical example of microneedle had been constructed utilizing hyaluronic acid integrated with stretch triggered drug nanocapsules loaded micro-depot particles [219]. This wearable device is designed on a thin elastomeric material that can be worn and deliver nanocapsules triggered by body motion. Alternatively, a finger-powered microfluidic-based system was used to induce the flow of drug release toward the microneedle array. Whereas, the bendable microneedles possess the advantage of pumping might be connected with the microfluidic network. Both the system was capable of delivering the drug under the influence of external stimuli as pressure and structural deformation. In 2017, Lee et al. [219] proposed an alternative approach to control drug delivery using temperature-responsive microneedles. Herein, the proposed wearable patch and a disposable strip-based sweat sensor integrated with glucose and physiological parameters sensing which control the actuation of the microneedles. The microneedle was thermally actuated from a loop-system based on the correlation of measured signals. This device exhibited promising results while tested in diabetic mice and after delivering the drug through microneedle showed a decrease in glucose level in blood sample upon thermal actuation. Considering ISF an important bio-fluid is highly advantageous in wearable biosensors due to ease of correlation of metabolites concentration to blood plasma levels and require minimal invasive method [199]. Similar to other biofluids (saliva, sweat, tears), commercial aspects of ISF-based wearable biosensors is still not achieved full potential. Even a well-known example of ISF-based biosensor, i.e., GlucoWatch Biographer (Cignus, Inc.) [213] has not explored to its full potential and withdrawn due to sampling errors, frequent calibration and irritation to skin. FreeStyeLibre, a wearable device for continuous monitoring of glucose, was launched by Abbott with an easy and user-friendly interface [17,171]. This device is integrated with a small size sensory part and can be worn on the upper arm. Without any finger-stick calibration, it could stand for e period of 14 days. The data can easily be transferred wirelessly via a near infra-red identification tag.

A first successful example of an implantable biosensor was reported by Iost et al. [223] to monitor blood glucose intravenously in rats. In the design, the microelectrode is made-up of a carbon-based composite of flexible carbon fibers, neutral red as a mediator and GOx as an enzyme. A flexible biochip of 100 mm was prepared using a long polypropylene catheter to fabricate electrodes, and these fabricated FCF-based microelectrodes were pre-treated with concentrated nitric acid, and every single fiber was separated with microscopy tweezers. For practical utility, the authors characterized the performance of the device for in vivo glucose monitoring via simulating the diabetic model by injecting glucose to rat blood. Yang et al. (2017) developed a flexible self-powered implantable electronic-skin as a real-time diagnose of kidney health by in-situ monitoring of urea/uric-acid based on the piezo-enzymatic-reaction [224] on the surface of ZnO nanowire (Figure 8). Herein, the reaction between the clinical analyte, i.e., urea/uric acid and enzyme (urease/uricase) immobilized over ZnO nanowires surface to increase the surface carrier density of ZnONW, thereby increasing the piezo-electric impulse [225]. In pure water, the ZnO nanowires do not respond to the piezoelectric output signal (Figure 8a), whereas deformed ZnO nanowires (Figure 8b) produce a piezoelectric signal [226]. As no chemical reaction occurs on the nanowire surface, the surface carrier density is very low that leads to the generation of high piezoelectric output. In addition to uric acid, the enzymatic reaction on the surface of the enzyme@ZnO nanowires results in a piezoelectric output which corresponds to analyte concentration (Figure 8c). In this arrangement, H^+^ and e^−^ adsorbed on the surface of ZnO and reinforce the piezo effect. On applying the force (Figure 8d), the nanowires generate lower piezo output. Figure 8e depicts the enzymatic events taking place in the generation of charge. This piezoelectric-biosensing process does not require an external electricity supply and a successful example of self-powered in situ body fluids-analysis mainly for a patient suffering from chronic diseases.

Aptamers are the single-stranded short sequences of DNA or RNA oligonucleotides, which are obtained via an in vitro process known as SELEX (Systemic Evolution of Ligands by Exponential Enrichment) [227]. These biomolecules offer advantages of higher selectivity, sensitivity, thermal stability, etc., which make them an ideal candidate for target analyte detection [8,228]. Based on aptamer-functionalized graphene, a stretchable and ultra-flexible field-effective transistor-based nanosensor was reported by Wang et al. [229]. The binding event of aptamer and target analyte and aptamer induces a change in the graphene carrier concentration corresponds to the analyte concentration. Based on a thickness of 2.5-µm Mylar substrate, this nanosensor was capable of conforming to the underlying surfaces that undergo structural deformations. Implying the large structural deformations applied cyclically or non-recurrently, this device allows the detection of immune response biomarkers, i.e., TNF-α, cytokines with higher selectivity and a very low limit of detection down up to 5 × 10^−12^ M.

Implantable WBs possesses various advantages over other wearable devices in monitoring the metabolites, nerve impulse, body parameters and drug delivery [230]. For example, blood pressure monitoring being a critical parameter, reflect the injury or damage to the physiological functions. Various implantable and clinically proven miniaturized devices for blood pressure are continuous efforts in monitoring patient health [231]. While academic research reaches some initial level, considerable attention and success in the field of implantable wearables have been achieved [4,232], and the results have started to appear in the commercial market [108].

Earlier in 2005, the first implantable biosensor for continuous monitoring of glucose was launched by Medtronic (USA) [233]. This self-implanted glucose biosensor was catalyzed by amperometric approaches based on the H_2_O_2_-oxidation by the glucose in the presence of GOx. This device collects the data every 5 min time for 3 days. Whereas, the glucose biosensor launched by Dexcom (USA) having a longer-life of 7 days [17]. The Sensors for Science and Medicine knew as Senseonics introduced a wholly implantable glucose biosensor for in vivo glucose monitoring worn in the upper arm, which lasts up to 29 days [234]. This design working principle is based on the chemistry of anthracene-derived diboronic acid established by James et al. [235]. Design integration involved an introduction of a novel strategy that prevents the breakdown of boronic acid-based recognition coupled with a fluorescent transduction molecule. Additionally, a Pt layer was used to decompose hydrogen peroxide (H_2_O_2_) produced via inflammatory response and powered via an RF power approach through an inductive coupling by an external induction coil. This creates a battery-free sensor platform.

Even after the medical and commercial successes, there are several lacunas in these devices, as a finger stick blood test is still required to be performed while using the self-implantable biosensor devices. Currently, there is still a lot of work necessary to establish the reliability, stability, and biocompatibility of the material selected for developing wearable, implantable biosensor having longer shelf-life (up to a few months or year) and wider utilities [213].

## 4. Outlook and Future Perspectives

The correct estimation of determinants of human health and biological markers to describe the health condition of humans is highly important. Since, the emergence of lab-on-chip analytical devices, the development of health monitoring devices and wearable bioelectronics came into existence with human skin and tissue interfaces. In order to design a flexible wearable device, it is promisingly important to improve the safety, stability and reliability of the test device. Simultaneously there is room for improvement of clinical aspects and analytical detection in case of chronic disease and its continuous real-time health monitoring. The need for fast, robust and reliable wearable devices with ease of worn is also recommended. In the present review, we discussed, in detail, wearable sensors, their types with their inherent characteristics, especially focused on the wearable based design and utility, bio-multifunctionality (self-healing, biodegradable, biocompatibility, etc.) of sensors and detection principle (optical, electrochemical) used. Furthermore, the various wearable biosensors for monitoring of biological marker (IL, cytokinin), metabolites (glucose, lactate) and physiological parameters (pH) were discussed in detail. Various applications have been discussed, whereas, there is a still need for high precision, biocompatibility, higher sensitivity, accuracy, robustness, and stability of wearable devices.

As the current status of functionalities of wearable and flexible sensors is increasing, the development of novel wearable devices which can fill the gap and offer the advantage in human health monitoring and medical application is evolving. The availability of portable biosensing devices to determine the analyte in bio-fluids for early detection of human health has gained considerable attention. Screening of blood is still the most important bio-fluid for individual health monitoring; however, much more interest has been shifted towards other naturally secreted body-fluids having a composition similar to the blood and easily accessible. A direct correlation between the analyte of interest and biological fluids can be established. In this regard, body-fluid mainly saliva, tears, sweat are considered due to the availability of non-invasive techniques to collect and analyze samples.

Despite considerable progress and development in wearable biosensors/devices, the current state of the art of wearable biosensing swim around demonstrating the proof-of-concept of wearable devices for the determination of various biomarkers. However, very few steps have been taken toward practical applications and commercialization in this field. The parameters considered are stability, precision, accuracy, stability, communication, etc., the wearable biosensors encountered many fundamental challenges and gaps in technology related to the scope. Overcoming these challenges is critical to successful growth and widespread commercialization.

Microfluidic integration and advancements in biosensor miniaturization have tremendously transformed the biosensors concerning their sensitivity, robustness, stability and portability. In the future, the architect and construction of novel wearable devices will rely on the various abovementioned parameters instead of single ones. The colorimetric biosensor offers the advantage of visual detection and an integrated mobile-based readout system. Moreover, the higher sensitivity comes with electrochemical techniques. On the other hand, microfluidics or flow-based systems will be important factors in the near future for real-time or continuous monitoring of the analyte of interest. In the present review, we discussed the importance of microfluidic-based wearable sensors with improved performance. For a practical approach, the collection of samples and minimization of sample volume and specification of the delivery of collected samples such as sweat, saliva, ISF, and tears are important towards the active sensing region of the sensor. All of these will provide characteristics of microfluidic platforms that will be an addition to wearable biosensor technology.

Various advanced and innovative wearable biosensing strategies/devices have been reported for the detection of a wide range of analytes such as ions (Na^+^, K^+^, NH_4_^+^, Ca^2+^), human metabolite (lactate, glucose) in various biological fluids. There resemble the blood utilizing enzymatic and non-enzymatic catalyzation. Improvements in biofouling, sampling strategies, availability of flexible and biodegradable material and wireless communications have benefitted the wearable technology, which indirectly improves the reliability of these sensors, capability of analyte monitoring and wearability.

Nevertheless, we believe in the potential of technology and wait for innovative experimentation and approaches with enhanced efficacy. With the improvements and globalization trends in the biosensor market, paradigm-changing technologies always come quickly and grab attention. In the coming future, if these recent trends reach commercial success, it would be a vital achievement in the healthcare industry and patient life.

## Figures and Tables

**Figure 1 molecules-26-00748-f001:**
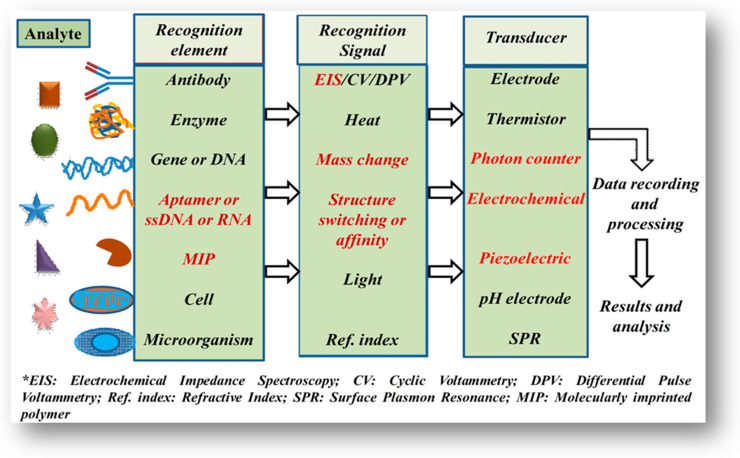
Schematic representation of biosensor components (illustrated from Sharma et al. 2017; Ref [8]).

**Figure 2 molecules-26-00748-f002:**
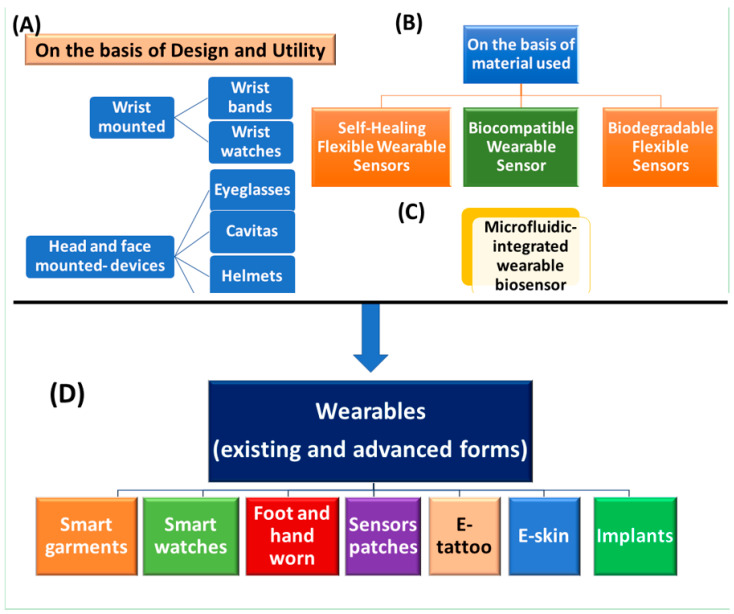
Classification and different forms of wearable devices and accessories (**A**) on the basis of design and utility; (**B**) on the basis of material employed in the construction of wearables biosensor; (**C**) on the basis of transduction principle; (**D**) existing and advanced forms of wearable devices.

**Figure 3 molecules-26-00748-f003:**
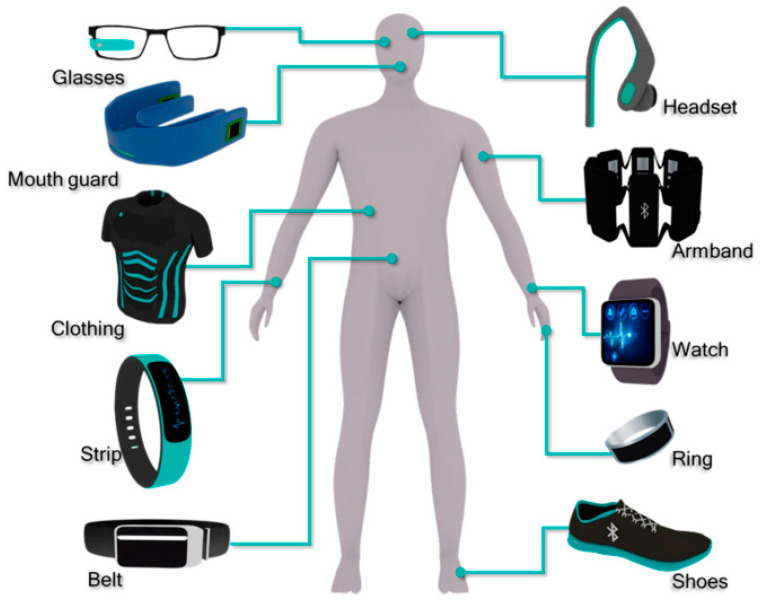
Portable and healthcare devices worn on body parts (reprint adapted with permission from Guk et al., 2019@MDPI; Ref [41]. Copyright 2019 MDPI).

**Figure 4 molecules-26-00748-f004:**
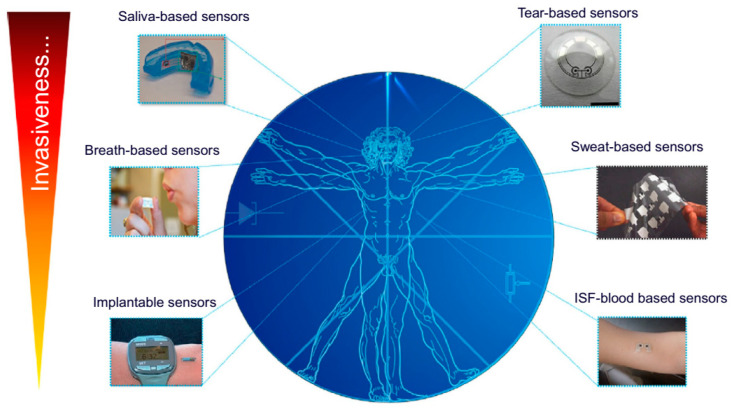
Overview of the rapidly growing field of wearable biosensors (reprint with permission from Pappa A M et al., 2018; Ref. [170]. Copyright 2018 Elsevier Ltd.).

**Figure 5 molecules-26-00748-f005:**
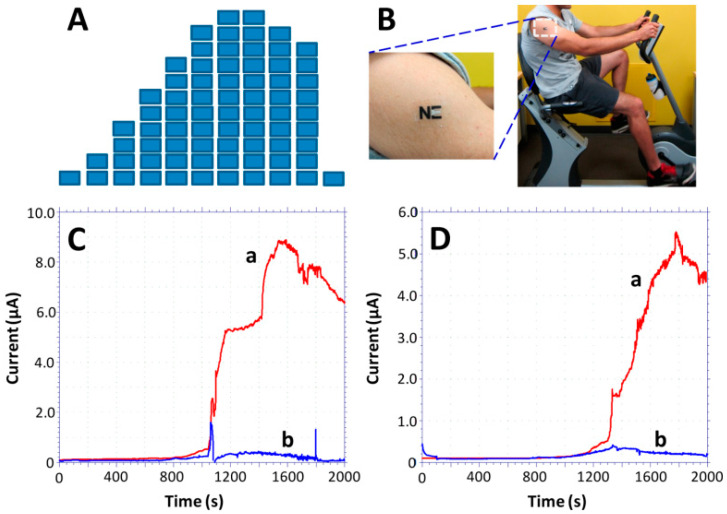
Monitoring of sweat lactate during 33 min of cycling exercise while changing the work intensity. (**A**) Exercise resistance profile on a stationary cycle. Subjects were asked to maintain a constant cycling rate while the resistance was increased every 3 min for a total evaluation of 30 min. A 3-min cool-down period followed the exercise. (**B**) An “NE” lactate biosensor applied to a male volunteer’s deltoid; (**C**,**D**) Response of the LOx- (a) and enzyme-free (b) tattoo biosensors during the exercise regimen (shown in part **A**) using two representative subjects. Constant potential, +0.05 V (vs. Ag/AgCl); measurement intervals, 1 s (reprint with permission from Jia et al. 2013; Ref. [203]. Copyright 2013 American Chemical Society).

**Figure 6 molecules-26-00748-f006:**
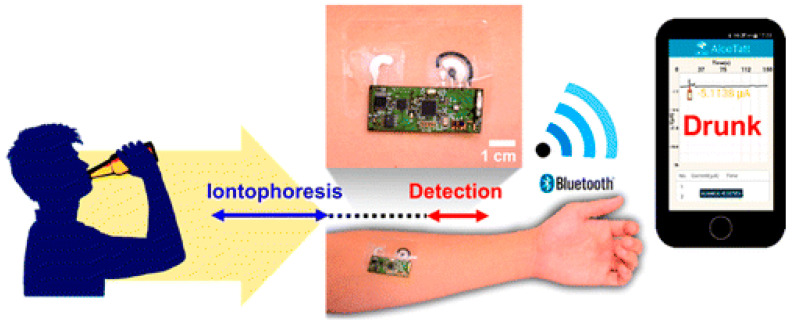
Schematic diagram of a wireless operation of the iontophoretic-sensing tattoo device for transdermal alcohol sensing. In the diagrams of the tattoo-base device, blue and red highlights show the active zones during iontophoresis and amperometric detection, respectively (reprint adapted with permission from Kim et al. 2016; Ref. [168]. Copyright 2016 American Chemical Society).

**Figure 7 molecules-26-00748-f007:**
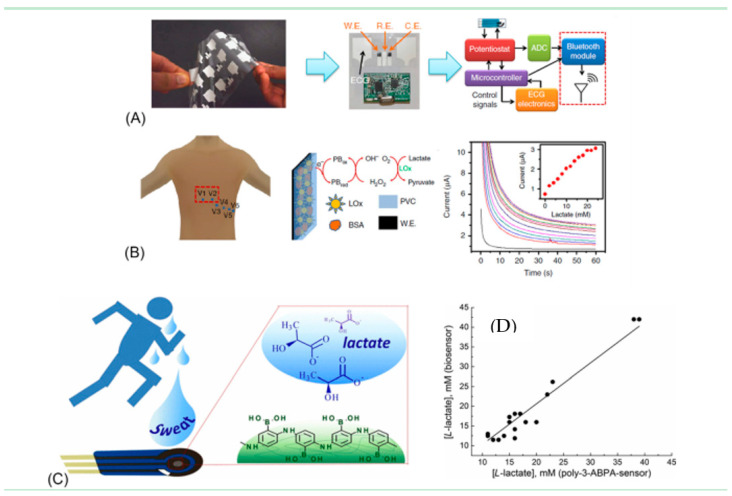
Device photographs of sweat-based sensor on body and its sensor features (**A**,**B**), schematic representations of screen- print, block diagram of the circuit, enzymatic sensing mechanism and amperometric response of wearable sweat-based lactate biosensor (**C**), schematic illustration of nonenzymatic lactate sensor in human sweat and correlation diagram for lactate measurement in 17 sweat samples collected from seven healthy human subjects measured using poly(3-APBA) sensor and with the flow-injection system equipped# with the biosensor (**D**) (reprint adapted with permission from Parlak et al., 2020; Ref. [17]. Copyright 2020 Elsevier Ltd.).

**Figure 8 molecules-26-00748-f008:**
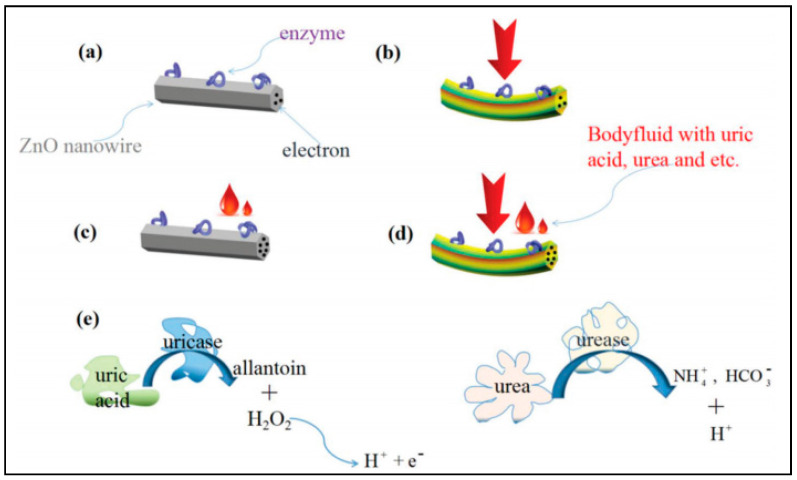
The working mechanism of a self-powered implantable e-skin. (**a**) Enzyme@ZnO nanowire in pure water without a compressive force. (**b**) The piezoelectric output of the enzyme@ZnO nanowire in pure water under an applied force. (**c**) Enzyme@ZnO nanowire in the corresponding solution without a compressive force. (**d**) The piezoelectric output of the enzyme@ZnO nanowire in a corresponding solution under an applied force. (**e**) The enzymatic reactions on the surface of the nanowire (reprint adapted with permission from Yang W et al., 2018; Ref. [224]. Copyright 2018 RSC Publisher).

## Data Availability

Data available in a publicly accessible repository.

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
