# Peer review of "Wearable Biosensors: An Alternative and Practical Approach in Healthcare and Disease Monitoring"

_molecules, 2021, doi:10.3390/molecules26030748_

Round 1
Reviewer 1 Report
The manuscript titled "Wearable Biosensors: An Alternative and Practical Approach in Healthcare and Disease Monitoring" tries to compile all the advances made in the field of wearable biosensors. The manuscript has been written with an extensive amount of literature survey. However, there are a few points that needs to be addressed before publication.
- It would be beneficial for a common reader if the authors included the pros and cons of each system and what methods could be employed to circumvent those problems. since reviews are not merely a compilation of available literature but an in depth study of that field of work and a valid discussion of the caveats , advantages as well as possible directions of future work.
- The authors should discuss the different modes of detection used in wearable sensors and discuss them in detail instead of just mentioning them. Readers are interested in how a smartwatch works and not just which one is measures which biomarker.
- The authors should refrain from using language directly from the papers they are quoting and instead should come up with their own summary.
- The authors could minimize the size of the manuscript by avoiding repetition of sentences in the different sections.
- The manuscript needs extensive English grammar and language check. some of the sentences for example like "Scientists and lab workers can be precisely measured biomarkers in biological fluids such as sweat, saliva, tears, interstitial fluid, blood, urine in order to monitor health condition and metabolism utilizing wearable biochemical sensors" does not make any sense. This is just one example, there are many instances where the sentences do not make any sense or are confusing.
Author Response
The manuscript titled "Wearable Biosensors: An Alternative and Practical Approach in Healthcare and Disease Monitoring" tries to compile all the advances made in the field of wearable biosensors. The manuscript has been written with an extensive amount of literature survey. However, there are a few points that needs to be addressed before publication.
Thank you so much for your valuable comments and suggestion.
- It would be beneficial for a common reader if the authors included the pros and cons of each system and what methods could be employed to circumvent those problems. since reviews are not merely a compilation of available literature but an in-depth study of that field of work and a valid discussion of the caveats, advantages as well as possible directions of future work.
Response: We thank the reviewer for beneficial comments. Unfortunately, there is no specific section for the advantage and disadvantage of various methods to avoid broad extension of manuscript. However, important limitation, merits and demerits of various system have been briefed throughout the review as highlighted in red (such Page 18 and 30).
- The authors should discuss the different modes of detection used in wearable sensors and discuss them in detail instead of just mentioning them. Readers are interested in how a smartwatch works and not just which one is measures which biomarker.
Response: The authors appreciate the valuable suggestions from learned Reviewer. We have already discussed different mode of detection (kindly refer section 2.3.1 and 2.3.2) but in the revised manuscript we have added some comprehensive discussion in some of the techniques as highlighted in red. However, to avoid the lengthiness of manuscript exhaustive discussions have been avoided.
As suggested, the basic principle behind the working of wrist mounted sensors such as smartwatch has been added in the concerned section (Page 8, Line 171-176).
- The authors should refrain from using language directly from the papers they are quoting and instead should come up with their own summary.
Response: As per the suggestion, we have thoroughly checked the entire manuscript to omit the overlapping/ or similar sentences in various sections such as at Page 8 (Line- 191-195).
- The authors could minimize the size of the manuscript by avoiding repetition of sentences in the different sections.
Response: As suggested, we have gone through the manuscripts and possible repetition of sentences have been avoided or removed at appropriate places in the revised manuscript.
- The manuscript needs extensive English grammar and language check. some of the sentences for example like "Scientists and lab workers can be precisely measured biomarkers in biological fluids such as sweat, saliva, tears, interstitial fluid, blood, urine in order to monitor health condition and metabolism utilizing wearable biochemical sensors" does not make any sense. This is just one example, there are many instances where the sentences do not make any sense or are confusing.
Response: We apologize for the language errors. The review has been re-read, and re-corrected to remove the possible grammatical mistake and necessary correction has been incorporated to improve quality and wider readability.
imethylsiloxane) and PU(polyurethane) as an example in 2.2.2. Biocompatible Wearable Sensor. Because these materials are highly biocompatible polymer, therefore, widely used in wearable sensors.
Response: As per suggestion the examples of PDMS and PU have been added in the revised manuscript at page 20 (Line 468-469).
Reviewer 2 Report
1. In the material, there are no extra explanations of Fig. 7a, 7b, 7c, 8b, 8d, 8e.
2. Background explanation in Abstract is too long.
3. It would be nice to add the chest-mounted devices in 2.1. On the basis of their design or utility.
4. It would be nice to add the wrist patches as an example in 2.1.1. Wrist-Mounted Wearable.
5. It would be nice to add the self-adhesive flexible wearable sensors in 2.2. Bio-Multifunctional Smart Wearable Sensors.
6. It would be nice to add the hydrogel as an example in 2.2.1. Self-Healing Flexible Wearable Sensors. Because it is widely used for self-healing material.
7. It would be nice to add the PDMS(polydimethylsiloxane) and PU(polyurethane) as an example in 2.2.2. Biocompatible Wearable Sensor. Because these materials are highly biocompatible polymer, therefore, widely used in wearable sensors.
Author Response
- In the material, there are no extra explanations of Fig. 7a, 7b, 7c, 8b, 8d, 8e.
Response: The detailed explanations of the Fig. 7a, 7b, 7c at page 34-35 (Line 819-835) and for 8b, 8d, 8e at page 39 (Line 927- 932) has been added.
- Background explanation in Abstract is too long.
Response: As per suggestion, the abstract has been re-written in the revised manuscript accordingly.
- It would be nice to add the chest-mounted devices in 2.1. On the basis of their design or utility
Response: As suggested, a paragraph on chest-mounted device has been added under design and utility section at page 15-16 (Line 360-371) with appropriate references (Ref 94-98).
- It would be nice to add the wrist patches as an example in 2.1.1. Wrist-Mounted Wearable
Response: A paragraph on wrist patches has been added under section 2.1.1. wrist-mounted wearable at page 9-10 (Line- 218-243) with suitable references (Ref 57-61).
- It would be nice to add the self-adhesive flexible wearable sensors in 2.2. Bio-Multifunctional Smart Wearable Sensors.
We thank the reviewer for their valuable advice. However, author apologize that the above mentioned could not be added to limit the extensive lengthening of the review.
- It would be nice to add the hydrogel as an example in 2.2.1. Self-Healing Flexible Wearable Sensors. Because it is widely used for self-healing material.
Response: As suggested, a section on hydrogel under section of self-healing flexible wearable sensors has been added at page 17-19 (Line- 411-428) with appropriate references (Ref 114-117) in the revised manuscript.
- It would be nice to add the PDMS(polydimethylsiloxane) and PU(polyurethane) as an example in 2.2.2. Biocompatible Wearable Sensor. Because these materials are highly biocompatible polymer, therefore, widely used in wearable sensors.
Response: As per suggestion the examples of PDMS and PU have been added in the revised manuscript at page 20 (Line 468-469).
Round 2
Reviewer 1 Report
The authors are strongly requested to use professional help for correcting the language of the manuscript.